# RAF1 as a standalone therapeutic target in KRAS-driven lung adenocarcinoma: No added efficacy from co-targeting ARAF, EGFR, or DDR1

Laura de-la-Puente-Ovejero[1,2☯], Ana Fernández-Rodríguez[1,2☯], Sarah Francoz[1], Gonzalo Aizpurua[1], Lucía Lomba-Riego[1], Matthias Drosten[2,3], Carmen Guerra[1,2], Mónica Musteanu[2,4,5*], Mariano Barbacid[1,2*], Sara García-Alonso[1,2*]

**1** Experimental Oncology Group, Tumor Biology Programme, Centro Nacional de Investigaciones Oncológicas (CNIO), Madrid, Spain, **2** Centro de Investigación Biomédica en Red de Cáncer (CIBERONC), Instituto de Salud Carlos III, Madrid, Spain, **3** RAS Signalling and Lung Cancer Group, Molecular Mechanisms of Cancer Program, Centro de Investigación del Cáncer, Consejo Superior de Investigaciones Científicas-Universidad de Salamanca, Salamanca, Spain, **4** Department of Biochemistry and Molecular Biology, Faculty of Pharmacy, Universidad Complutense de Madrid, Madrid, Spain, **5** Instituto de Investigación Sanitaria del Hospital Clínico San Carlos (IdISSC), Madrid, Spain

☯ These authors contributed equally to this work.
* sgarciaa@cnio.es (SG-A); mbarbacid@cnio.es (MB); mmustean@ucm.es (MM)

## Abstract

### Background/objectives

KRAS-mutant lung adenocarcinoma remains without effective targeted therapies for most patients, particularly those with non-G12C alleles or resistance to KRAS[G12C] inhibitors. RAF1 is essential for KRAS-driven tumor maintenance through kinase-independent survival functions, making it an attractive candidate for targeted protein degradation. However, the therapeutic impact and safety of co-targeting RAF1 with related kinases remain unclear.

### Methods

We used dual-recombinase genetically engineered mouse models of *Kras[+/G12V];Trp53[-/-]* lung cancer to evaluate the effects of *Raf1* ablation alone or in combination with *Araf*, *Egfr*, or *Ddr1*. Lung tumors were initiated by intranasal Ad5-CMV-FLPo delivery and allowed to reach CT-detectable size before inducing systemic gene deletion via tamoxifen-activated CreERT2. Tumor burden was monitored by longitudinal CT imaging and classified using RECIST-like criteria. Toxicity was assessed by body weight monitoring, histopathology of major organs, and survival analysis.

### Results

*Raf1* deletion induced robust tumor regression within two months, in more than 60% of lesions. *Araf* ablation alone or combined with *Raf1* did not affect tumor initiation,

**Data availability statement:** All relevant data for this study are within the paper and its Supporting information files.

**Funding:** This study was financially supported by the Agencia Estatal de Investigación, Ministerio de Ciencia, Innovación y Universidades (MICIU/AEI/10.13039/501100011033) (https://www.aei.gob.es) in the form of a grant and predoctoral fellowship awards received by Ld-I-P-O (FPU21/04678) and AF-R (FPU22/02924). This study was also financially supported by the European Regional Development Fund (https://ec.europa.eu/regional_policy/funding/erdf_en) - "A way of making Europe" in the form of grants received by MB and MM (RTI2018-094664-B-I00) and MM (PID2021-122797OB-I00). This study was also financially supported by the Autonomous Community of Madrid (https://www.comunidad.madrid) in the form of a grant (S2022/BMD-7437, iLUNG 2.0-CM) received by MB. This study was also financially supported by the CRIS Cancer Foundation (https://criscancer.org.uk) in the form of an award to MB. This study was also financially supported by the AXA Research Fund (https://axa-research.org) in the form of an Endowed Chair award received by MB. This study was also financially supported by the a CIBERONC Fund (https://ciberonc.es) in the form of a grant (CB21/12/00121) received by MB. This study was also financially supported by AECC Talent (https://aecctalent.com) in the form of a Postdoctoral Fellowship award (2020-2022) received by SG-A. This study was also financially supported by the Juan de la Cierva Investigadores Fellowship in the form of an award (FJC2018-036013-I) received by SG-A. This study was also financially supported by the Agencia Estatal de Investigación, Ministerio de Ciencia, Innovación y Universidades (MICIU/AEI/10.13039/501100011033) (https://www.aei.gob.es) in the form of a Ramón y Cajal Fellowship award (RYC2018-025415-I) received by MM. This study was also financially supported by the European Social Fund (ESF) (https://european-social-fund-plus.ec.europa.eu) - "Investing in your future" in the form of a grant. This study was also financially supported by "la Caixa" Foundation (https://lacaixafoundation.org) in the form of fellowship awards

progression, or regression rates. Similarly, neither genetic nor pharmacological EGFR inhibition (afatinib) improved responses to *Raf1* ablation. *Ddr1* co-deletion also failed to enhance therapeutic efficacy and slightly reduced response rates. None of the dual-targeting strategies increased systemic toxicity.

## Conclusions

RAF1 is a key, non-redundant vulnerability in KRAS-driven lung adenocarcinoma. Co-targeting ARAF, EGFR, or DDR1 provides no additional therapeutic benefit in established disease. The absence of adverse effects from ARAF co-deletion suggests that RAF1 degraders with partial cross-activity towards ARAF are likely to be safe. These findings provide a strong preclinical rationale for developing RAF1-targeted degradation as a monotherapy for these malignancies.

---

## Introduction

Lung cancer remains the leading cause of cancer-related mortality worldwide, accounting for nearly 1.8 million deaths annually. Non-small cell lung cancer (NSCLC) represents approximately 85% of cases, with lung adenocarcinoma (LUAD) being the most common subtype [1]. A significant proportion of LUAD is driven by oncogenic mutations in KRAS, particularly affecting codon 12, which lead to constitutive activation of downstream signaling pathways that promote tumor proliferation, survival, and metastasis [2]. Despite recent advances in KRAS[G12C]-selective inhibitors [3,4], most KRAS-mutant lung cancers lack effective targeted therapies, highlighting the need for alternative strategies that disrupt essential signaling nodes within the RAS effector network.

RAF kinases (ARAF, BRAF, and RAF1/CRAF) are central mediators of RAS signaling via the RAF/MEK/ERK pathway [5]. Genetic studies using mouse models have demonstrated that RAF1 plays a unique role in KRAS-driven LUAD [6]. Its deletion results in marked tumor regression without causing major systemic toxicity, largely due to its kinase-independent functions in suppressing apoptosis [7]. This has positioned RAF1 as an attractive therapeutic target, especially for emerging targeted protein degradation (TPD) approaches that can fully eliminate its oncogenic scaffold functions [8].

Degraders against RAF1 could disrupt both its kinase-dependent and non-catalytic functions, potentially leading to more durable clinical responses than traditional inhibitors. However, developing highly selective RAF1 degraders is challenging due to the high structural and sequence similarity within the RAF family [9]. Given that next-generation pan-RAF inhibitors or degraders may target multiple isoforms simultaneously, it is important to determine whether the combined loss of RAF1 and related kinases offers additional therapeutic benefit or introduces toxicity.

ARAF is the least understood member of the RAF family, displaying lower kinase activity than BRAF and RAF1 and context-dependent roles in cell survival and signaling [10,11]. While combined deletion of RAF1 and BRAF in KRAS-driven LUAD models did not improve tumor outcome and was poorly tolerated [6,12], the potential

received by GA (LCF/BQ/DR22/11950011) and
LL-R (LCF/BQ/DR23/12000028). The funders
had no role in study design, data collection and
analysis, decision to publish, or preparation of
the manuscript.

**Competing interests:** The authors have
declared that no competing interests exist.

contribution of ARAF has not been systematically addressed. Understanding whether
ARAF plays a supportive role in tumor maintenance is clinically relevant, as pan-RAF
degraders could potentially eliminate both isoforms in patients.

To explore additional cooperating vulnerabilities, we also examined EGFR and
DDR1 as candidate co-targets. Prior work in pancreatic ductal adenocarcinoma
(PDAC) models showed that combined ablation of RAF1 and EGFR induced complete tumor regression, whereas individual deletions were insufficient [13]. Similarly,
DDR1, a collagen receptor tyrosine kinase, was identified as a key mediator of
KRAS-mutant LUAD progression. Genetic deletion or pharmacological inhibition of
DDR1 blocked tumor initiation and suppressed progression in preclinical models [14].
These findings suggested that dual targeting of RAF1 with either EGFR or DDR1
could potentially enhance therapeutic efficacy in LUAD.

Here, we address these questions using genetically engineered mouse models (GEMMs) that allow temporal and spatial separation of tumor initiation and
target ablation. We first assessed whether ARAF deletion affects the initiation of
LUAD. We then investigated whether simultaneous deletion of RAF1 and related
kinases (ARAF, EGFR and DDR1) in established tumors driven by KRAS$^{G12V}$ and
loss of p53 enhances regression or induces toxicity compared to RAF1 deletion
alone.

Our results confirm that *Raf1* ablation alone is sufficient to induce marked tumor
regression and that co-deletion of *Araf*, *Egfr*, or *Ddr1* does not confer additional
therapeutic advantage. Importantly, systemic elimination of *Araf* caused no detectable adverse effects, suggesting that future RAF1-directed degraders with partial
cross-activity towards ARAF are unlikely to pose safety concerns. These findings
reinforce RAF1 as a key vulnerability in KRAS-mutant LUAD and provide a strong
preclinical rationale for the development of RAF1 degraders as therapeutic agents for
this aggressive disease.

## Materials and methods

### Mouse models and alleles

The following genetically engineered alleles were used for this study: *Kras*$^{LSLG12Vgeo}$
[15], *Kras*$^{FSFG12V}$ [6], *Trp53*$^F$ [16], *Tg.hUBC-CreERT2*$^T$ [17], *Araf*$^L$ (this work),
*Raf1*$^L$ [18], *Egfr1*$^L$ [19], *Ddr1*$^L$ (this work). Description of the alleles used in this work
can be found in S1 Table.

For the therapeutic approach studies the following compound mice were generated:
*Kras*$^{+/FSFG12V}$;*Trp53*$^{F/F}$;*hUBC-CreERT2*$^{+/T}$ and *Kras*$^{+/FSFG12V}$;*Trp53*$^{F/F}$;*hUBC-CreERT2*$^{+/T}$;
*Raf1*$^{L/L}$ to be compared with *Kras*$^{+/FSFG12V}$;*Trp53*$^{F/F}$;*hUBC-CreERT2*$^{+/T}$;*Raf1*$^{+/+}$;*Araf*$^{L/Y}$,
*Kras*$^{+/FSFG12V}$;*Trp53*$^{F/F}$;*hUBC-CreERT2*$^{+/T}$;*Raf1*$^{L/L}$;*Araf*$^{+/Y}$, *Kras*$^{+/FSFG12V}$;*Trp53*$^{F/F}$;
*hUBC-CreERT2*$^{+/T}$;*Raf1*$^{L/L}$;*Egfr*$^{L/L}$ and *Kras*$^{+/FSFG12V}$;*Trp53*$^{F/F}$;*hUBC-CreERT2*$^{+/T}$;
*Raf1*$^{L/L}$;*Ddr1*$^{L/L}$.

All mice used in this work were housed in specific-pathogen-free conditions in the
Animal Facility of the Spanish National Cancer Research Centre (AAALAC, JRS:
dpR 001659), in accordance with Federation of European Laboratory Animal Science

Association (FELASA) recommendations and following European Union legislation. Mice were subjected to 12-hours light/dark cycles in ventilated racks under controlled conditions of temperature and humidity. Animals were housed in groups of 4–5 per cage with environmental enrichment. Animals had access to sterilized tap water and chow ad libitum. They were fed with a standardized diet (or a tamoxifen diet (depending on the time point of the experiment). Both female and male mice were used for the experiments. Health status and body weight were monitored three times per week by the researchers and daily by the animal facility care-takers, ensuring early detection of any signs of distress or disease. A total of 245 animals were included in the study. 12 mice did not develop lung tumors and were sacrificed. From the remaining 233 mice, 8 were found dead in cage, most probably due to respiratory disfunction, and the rest were sacrificed at different times according to the experimental design: 10 mice at 6 months post-infection, 35 mice after 2 months of treatment (experiment duration 7–8 months) and 188 reaching predefined *humane endpoint*. Criteria for determining *humane endpoint* included loss of more than 20% of the initial body weight, abnormal activity, abnormal physical appearance (fur, skin, posture, etc.), and signs of respiratory malfunction (dyspnea, rales and extensive atelectasis detected by computerized tomography (CT) measurement). Once animals showed the above-mentioned symptoms, they were sacrificed within the following 10 minutes. Mice were euthanized by $CO_2$ inhalation according to the guidelines of the Ethics and Animal Welfare Committee (CEyBA).

All personnel involved in the study were adequately trained and certified in laboratory animal science, meeting the educational and competency standards recommended by FELASA to ensure responsible and ethical handling of animals throughout the research process. All experiments were approved by the Ethical Committees of the CNIO, the Carlos III Health Institute and the Autonomous Community of Madrid (PROEX 270.6/21 and PROEX 071.7/22).

## Necropsy and histopathology

Necropsies were carried out in the dissection laboratory at CNIO, involving the sacrifice of mice in a $CO_2$ chamber (5-min cycle). Death was confirmed by cervical dislocation before tissue harvesting. A comprehensive approach was followed to collect tissue samples, which were then subjected to various preservation methods based on experimental requirements. Specifically, samples were taken in 10% formalin, for subsequent inclusion in paraffin blocks, and cut and stained as needed by the Histopathology Unit at CNIO. Additionally, partial tissue samples were collected in microcentrifuge tubes and snap-frozen in dry ice-cooled 2-methylbutane for the subsequent extraction of DNA, RNA, or protein.

For histopathological visualization, tissue sections of 2.5 µm thickness were stained with Hematoxylin & Eosin (H&E). Alternatively, for quantification and classification of tumour lesions, lung lobes were processed for whole-mount X-Gal staining to detect β-geo, as a surrogate marker for $Kras^{G12V}$ expression. For immunohistochemistry (IHC) staining, a rat monoclonal α-RAF1 (1:200, Monoclonal Antibodies Unit, Ref. EMI411E), a rabbit monoclonal α-EGFR (1:200, Cell Signaling Technology Ref. 71655), and α-DDR1 (1:200, Cell Signaling Technology Ref. 5583) were used. Both H&E and IHC-stained sections were scanned using the Axio Scan.Z1 scanner (Zeiss). Visualization and analysis were performed with QuPath v0.5.1.

## Generation of *Araf$^{L/Y}$* mice

The targeting vector was generated by Gene Bridges GmbH (Heidelberg, Germany). Briefly, the final construct was obtained by inserting the loxP-FRT-PGK-gb2-neo/km-FRT-exon11-exon12-loxP cassette into a high-copy plasmid harboring the *Araf* genomic locus by Red/ET cloning. The vector was electroporated into G4 ES cells and recombinant clones were selected in the presence of G418 and ganciclovir. Recombinant clones were verified by Southern blot using genomic DNA digested with BamHI for both 3' and 5' arms and hybridized with external probes. Enzymatic digestion produced distinct bands: 15.7 kb (WT), 5.7 kb (recombinant 3'arm), and 9.4 kb (recombinant 5'arm) (S1 Fig). Three independent ES cell were microinjected into C57BL/6J blastocysts and transplanted into pseudo-pregnant

females. Male chimeras were bred with C57BL/6J females to obtain germline transmission. Oocytes from resultant females were fertilized with sperm from *Tg.CAG-Flp*[T/T] males. Complete cassette deletion was confirmed by genotyping PCR. All electroporation, microinjection, and genotyping procedures were carried out by the CNIO Transgenic Mice Unit.

### Generation of *Ddr1* lox mice

Sperm from the *Ddr1* targeted mutant strain B6NCrl;B6N-Atm1Brd *Ddr1*[tm1a(EUCOMM)Hmgu]/Orl (EM:09692; MGI:5318761) was purchased from the European Mouse Mutant Archive (Infrafrontier). To obtain a conditional floxed allele, where the exons 5 and 6 of the *Ddr1* gene are flanked by loxP sites, mice were crossed with *Tg.CAG-Flp*[T/T] transgenic animals to excise the FRT-flanked selection cassette from the *Ddr1*[tm1a(EUCOMM)Hmgu] targeting construct.

### Tumor induction and tamoxifen exposure

Lung adenocarcinomas were induced in 8- to 10-week-old anesthetized mice. Anesthesia was administered via an intraperitoneal injection of ketamine (75 mg/kg) and xylazine (12 mg/kg). Throughout the entire process, the eyes were protected with an ophthalmic ointment (Lacryvisc™) to prevent corneal drying. Tumors were initiated by intranasal instillation of a single dose of $10^6$ plaque-forming units (pfu) of adenovirus serotype 5 expressing optimized FLP recombinase under the CMV promoter (Ad5-CMV-FLPo). Activation of FLPo induced the expression of *Kras*[G12V] and loss of *Trp53*, leading to multifocal lung adenocarcinomas detectable by CT imaging within 5–6 months. For initiation studies, Ad5-CMV-Cre was used to directly activate the *Kras*[LSLG12Vgeo] allele.

Activation of the inducible CreERT2 recombinase was achieved by changing the standardized diet (2018S, Envigo) with a diet containing tamoxifen (TMX) (Teklad CRD TAM400 diet, Envigo) *ad libitum* for 8 weeks. The CreERT2 recombinase excised floxed alleles in tumor cells, leading to loss of target genes (S2 Fig). A diet transition to 2919S (Teklad) was introduced for one week after every two weeks on the TMX diet. Evidence of RAF1, DDR1, EGFR, and ARAF knockdown, confirming the model generation, is shown in S6 Fig.

### Tumor burden and response assessment

Tumor volumes were measured using a SuperArgus CompaCT scanner (Sedecal) under 4% isoflurane anesthesia (Braun Vetcare). Image processing, analysis and 3D rendering were performed using the 3D Slicer Viewer Software. Volumes were calculated assuming ellipsoid geometry:

$$V \approx (d_{long} \times d_{short} \times d_{short})/2$$

Mice were monitored daily for health status, and CT imaging was repeated every 4 weeks to quantify tumor volume changes. Mice were sacrificed at the end of the experiment or if they reached the *humane endpoint*—loss of more than 20% of the initial weight, anorexia, abnormal activity, abnormal physical appearance, and signs of respiratory malfunction. Kaplan-Meier survival curves were plotted for each cohort, and differences analyzed using log-rank tests.

Responses were categorized according to RECIST-like criteria [20]—Complete Regression (CR): disappearance of detectable lesion, Partial Regression (PR): ≥30% volume reduction, Stable Disease (SD): <30% reduction or ≤20% increase, Progressive Disease (PD): >20% increase.

### Toxicity evaluation

Potential systemic toxicity from dual target ablation was assessed by monitoring body weight throughout the study and performing necropsies at endpoint. Organs including heart, liver, spleen, kidney, and gastrointestinal tract were fixed, paraffin-embedded, sectioned (2.5 µm), and stained with hematoxylin and eosin (H&E) for histopathological evaluation.

Alternatively, for quantification and classification of tumor lesions, lung lobes were processed for whole-mount X-Gal staining to detect β-geo, as a surrogate marker for $Kras^{G12V}$ expression.

### Genotyping and validation of recombination

Tail or tissue biopsies were collected for genomic DNA extraction using high-salt lysis. The validation of correct allele excision was performed by the Genomics Unit at CNIO.

### Statistical analysis

Statistical comparisons between groups were made using unpaired Mann-Whitney test for non-normally distributed values. Kruskal-Wallis tests with post hoc corrections were applied for multiple comparisons. Survival analyses used log-rank (Mantel-Cox) tests. p-values <0.05 were considered significant. Analyses were conducted in GraphPad Prism v8.4.0 or R v4.4.2.

## Results

### Ablation of *Araf* does not influence tumor initiation or enhance the therapeutic effect of *Raf1* deletion in KRAS-driven LUAD

To assess whether *Araf* contributes to early tumorigenesis, the generated $Araf^{L/Y}$ allele (S1 Fig and Materials and methods) and the compound $Kras^{+/LSLG12Vgeo}$ mouse strain were used for this study. These mice were intranasally instilled with Ad5-CMV-Cre to simultaneously activate oncogenic *Kras* and excise *Araf*. Six months post-induction, mice developed multifocal LUAD lesions regardless of *Araf* status. Histological analyses revealed comparable numbers and sizes of tumors in *Araf*-deficient and control mice (Fig 1A). Quantification of total lung tumor burden showed no significant differences between groups (Fig 1B). These results indicate that ARAF is not required for the initiation of KRAS-driven LUAD.

To model therapeutic intervention, we used a dual-recombinase system allowing temporal separation of tumor initiation and gene ablation. $Kras^{+/FSFG12V};Trp53^{F/F}$ mice carrying floxed alleles for *Araf* and/or *Raf1* were infected intranasally with Ad5-CMV-FLPo to trigger oncogene activation and *Trp53* loss, leading to aggressive LUAD development. Tumor-bearing mice identified by CT imaging were subsequently exposed to a TMX-containing diet to activate the ubiquitously expressed CreERT2 recombinase, inducing systemic ablation of target genes (S2 Fig).

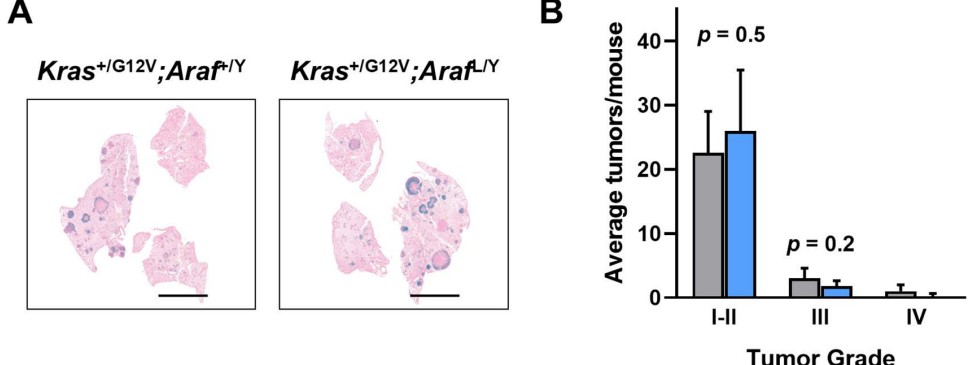

**Fig 1. Histological and quantitative analysis of LUAD initiation upon *Araf* ablation. (A)** Whole-mount X-Gal staining of lung sections from mice with the indicated genotypes, collected 6 months after AdCre treatment. β-Geo positive cells identified by X-Gal staining (dark blue) correspond to $Kras^{G12V}$ cells. Scale bars: 5 mm. **(B)** Number of tumors, classified by grade (**I–IV**), observed in $Kras^{+/G12V};Araf^{+/Y}$ (n = 5, grey bars) and $Kras^{+/G12V};Araf^{L/Y}$ (n = 5, blue bars) mice.

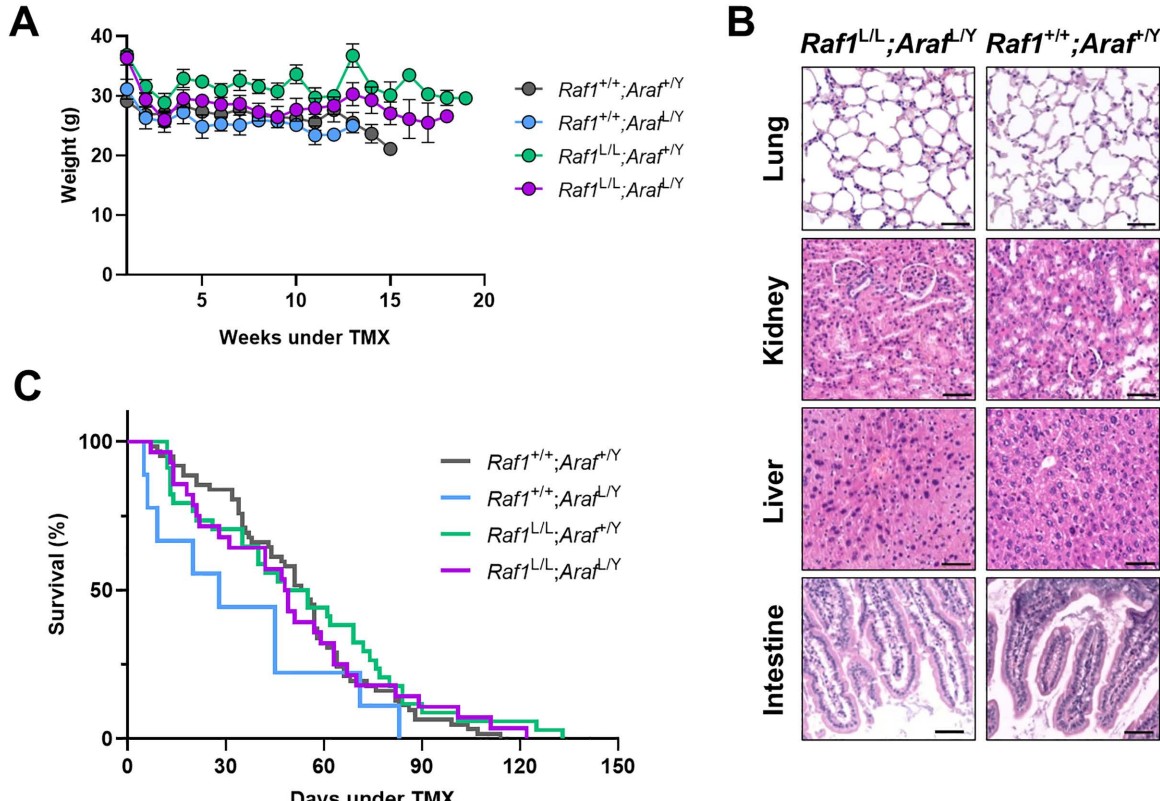

**Fig 2. *Araf* and *Raf1* dual ablation does not affect body weight, tissue integrity, or survival. (A)** Body weight measurements (in grams) of mice exposed to TMX-diet over time. Each line represents the mean body weight (± standard deviation) of the mice cohort at each timepoint. *Raf1*$^{+/+}$;*Araf*$^{+/Y}$ (n = 60, grey), *Raf1*$^{+/+}$;*Araf*$^{L/Y}$ (n = 8, blue), *Raf1*$^{L/L}$;*Araf*$^{+/Y}$ (n = 15, green), *Raf1*$^{L/L}$;*Araf*$^{L/Y}$ (n = 25, purple) **(B)** Histological analysis of H&E-stained tissues from dual-ablated and non-ablated mice. Scale bars: 50 μm. **(C)** Kaplan-Meier survival curves representing data from *Raf1*$^{+/+}$;*Araf*$^{+/Y}$ (n = 62, grey), *Raf1*$^{+/+}$;*Araf*$^{L/Y}$ (n = 9, blue), *Raf1*$^{L/L}$;*Araf*$^{+/Y}$ (n = 34, green), *Raf1*$^{L/L}$;*Araf*$^{L/Y}$ (n = 28, purple) mice.

Body weight monitoring revealed only a transient, diet-related decrease in all cohorts, with no significant long-term differences between controls and dual-ablated mice (Fig 2A). Histopathological examination of major organs, including lungs, liver, kidney, and intestine, showed no evidence of tissue damage or inflammatory responses attributable to target deletion (Fig 2B). Kaplan-Meier analysis demonstrated that both single and dual ablation of *Araf* and *Raf1* did not negatively impact survival compared with controls or *Raf1*-only ablation, respectively (Fig 2C). Together, these findings confirm that dual ablation of *Raf1* and *Araf* is well tolerated *in vivo* and does not cause systemic adverse effects.

We next evaluated therapeutic efficacy by tracking tumor burden in four experimental groups: (i) control, (ii) *Araf*-only ablation, (iii) *Raf1*-only ablation, and (iv) dual *Araf*/*Raf1* ablation. CT-positive lesions (≥5 mm3) were measured before and after 2 months of TMX treatment, and tumor response was classified according to RECIST criteria.

Consistent with previous studies, *Raf1* deletion markedly reduced tumor burden: 52.4% of lesions exhibited PR and 8.4% underwent CR (Fig 3A–3B). In contrast, *Araf* ablation alone had no significant effect on tumor growth, with most lesions continuing to progress similarly to controls. Simultaneous deletion of *Araf* and *Raf1* resulted in partial or complete regression in 67% of lesions. Interestingly, although a higher percentage of CR was observed, quantification of tumor volume as FC did not reveal a statistically significant reduction relative to *Raf1* deletion alone (Fig 3C). SD and PD frequencies were also comparable between the two groups.

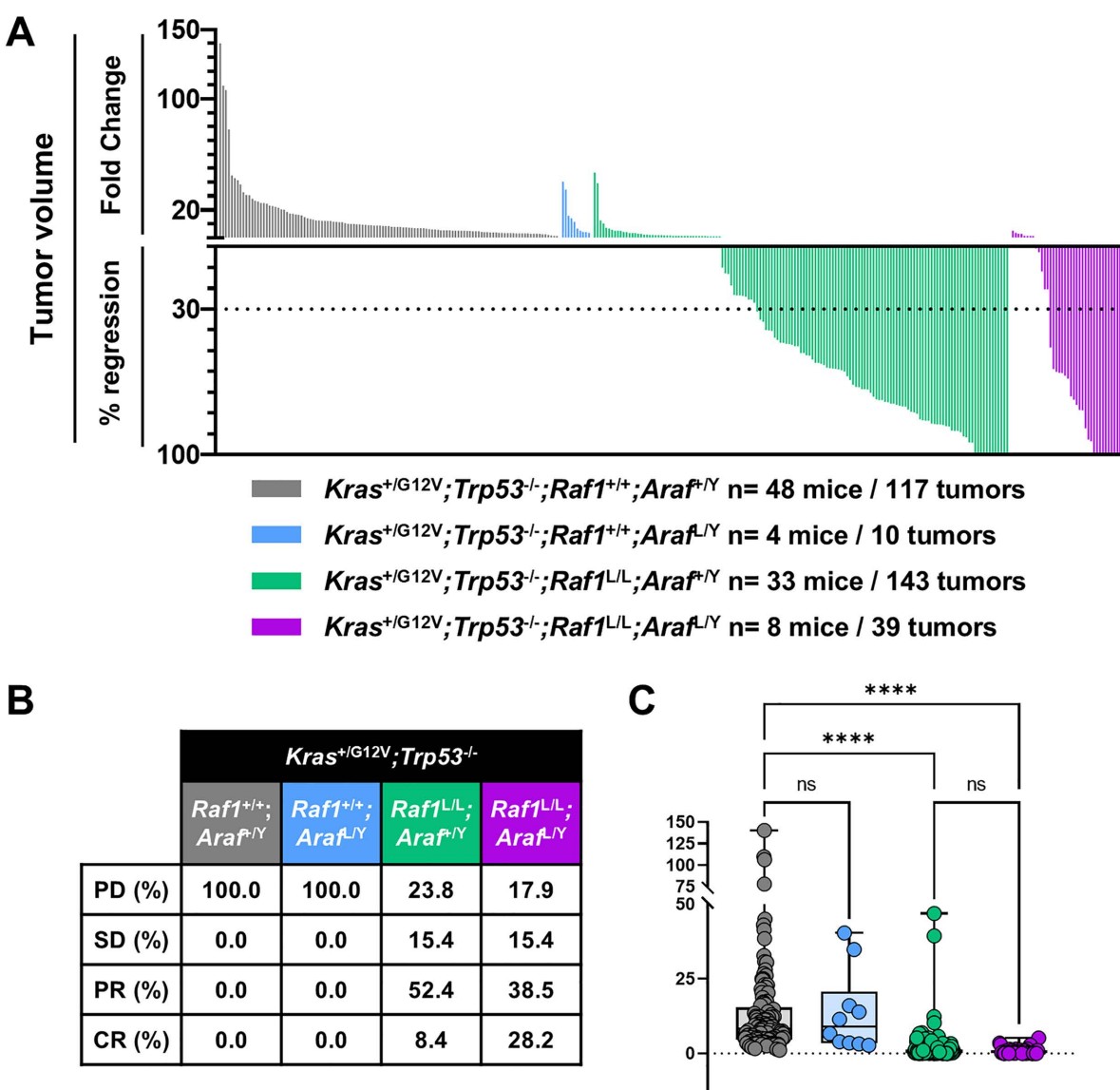

**Fig 3. Effect of *Araf* and *Raf1* genetic ablation on LUAD development.** (A) Waterfall plot representing the FC in tumor volume and the percentage of regression for individual CT-positive lung tumors in *Kras*<sup>+/FSFG12V</sup>;*Trp53*<sup>F/F</sup>;*hUBC-CreERT2*<sup>+/T</sup>;*Raf1*<sup>+/+</sup>;*Araf*<sup>+/Y</sup> (grey), *Kras*<sup>+/FSFG12V</sup>;*Trp53*<sup>F/F</sup>;*hUBC-CreERT2*<sup>+/T</sup>;*Raf1*<sup>+/+</sup>;*Araf*<sup>L/Y</sup> (blue), *Kras*<sup>+/FSFG12V</sup>;*Trp53*<sup>F/F</sup>;*hUBC-CreERT2*<sup>+/T</sup>;*Raf1*<sup>L/L</sup>;*Araf*<sup>+/Y</sup> (green), and *Kras*<sup>+/FSFG12V</sup>;*Trp53*<sup>F/F</sup>;*hUBC-CreERT2*<sup>+/T</sup>;*Raf1*<sup>L/L</sup>;*Araf*<sup>L/Y</sup> (purple) mice following 2 months of TMX diet. (B) Table indicating the percentage of tumors that show PD, SD, PR, and CR. (C) Statistical comparison of tumor volume FC between groups. $p$-values were obtained using the Kruskal-Wallis test with multiple comparisons. Data are presented as box-and-whisker plots showing the median (line), interquartile range (box), and minimum to maximum values (whiskers), with all individual data points displayed.

These data demonstrate that ARAF is dispensable for both LUAD initiation and progression in KRAS-driven mouse models. Most importantly, simultaneous ablation of *Araf* does not enhance the therapeutic benefit of *Raf1* deletion, indicating that RAF1 is a major vulnerability in this oncogenic context and that additional elimination of ARAF does not confer added efficacy.

## Concomitant targeting of *Raf1* and *Egfr* does not improve therapeutic outcome in KRAS-driven LUAD

To evaluate whether co-targeting an upstream receptor tyrosine kinase (RTK) could enhance the therapeutic effect of *Raf1* ablation, we tested the combination of systemic *Raf1* and *Egfr* deletion in *Kras*$^{+/G12V}$;*Trp53*$^{-/-}$ GEMMs. After tumor induction by Ad5-CMV-FLPo, mice harboring tumors greater than 5 mm3 were randomized to receive a TMX-containing diet for 60 days to induce systemic ablation of *Raf1* alone or in combination with *Egfr*. Tumor burden was monitored by CT imaging, and responses were classified according to RECIST criteria.

Body weight monitoring showed only a transient, diet-related decrease across all cohorts, with no significant long-term differences between controls and *Egfr* and *Raf1* dual-ablation. Kaplan–Meier analysis indicated that combined ablation of *Egfr* and *Raf1* did not adversely affect survival compared with controls (S3 Fig).

Simultaneous deletion of *Egfr* and *Raf1* did not further enhance therapeutic efficacy compared with *Raf1* deletion alone (Fig 4A–4B). Among 121 tumors analyzed, 43.8% displayed PR and 3.3% achieved CR, while 18.2% displayed SD and 34.7% remained growing. There were no statistically significant differences in overall tumor volume reduction or response category distribution compared to the *Raf1*-only group (Fig 4C).

To validate these findings using a pharmacological approach, we tested the effect of combining *Raf1* deletion with EGFR inhibition using the pan-ErbB tyrosine kinase inhibitor afatinib. Following tumor detection, mice received afatinib (10 mg/kg, 5 days per week) throughout the 60-day TMX treatment period. Tumor burden was monitored by CT and evaluated using the same response criteria.

The results closely mirrored those of the genetic model. The combination of *Raf1* deletion and afatinib treatment did not yield improved outcomes compared to *Raf1* ablation alone (S4 Fig). Response rates, including frequencies of PR, CR, and PD, were like those observed in the genetic cohorts. These findings reinforce the conclusion that EGFR inhibition, whether genetic or pharmacological, does not enhance the therapeutic efficacy of *Raf1* ablation in established KRAS-driven LUAD.

Together, these data suggest that LUAD tumors rely predominantly on RAF1-dependent signaling for maintenance and that upstream EGFR inhibition does not further impair tumor growth once RAF1 is eliminated. While previous studies in KRAS-driven PDAC models reported marked synergy between RAF1 and EGFR co-targeting [13], such cooperative dependency is not observed in the lung tumor context.

## Dual deletion of *Raf1* and *Ddr1* does not enhance tumor regression in KRAS-driven LUAD

DDR1 is a collagen-activated RTK implicated in the initiation and progression of KRAS-mutant LUAD. Prior studies have shown that genetic deletion or pharmacological inhibition of DDR1 impairs tumor development and growth in preclinical models of lung cancer [14]. To determine whether DDR1 contributes to tumor maintenance and whether its co-targeting could enhance the therapeutic response to *Raf1* ablation, we evaluated the effect of simultaneous deletion of *Ddr1* and *Raf1* in *Kras*$^{+/G12V}$;*Trp53*$^{F/F}$ GEMMs.

As in previous experiments, tumors were induced using Ad5-CMV-FLPo, and once CT-detectable lesions developed, mice were randomized to receive TMX for 60 days to induce systemic ablation of *Raf1* alone or in combination with *Ddr1*. Body weight remained stable across all cohorts, and Kaplan–Meier analysis showed no adverse effect of combined *Ddr1* and *Raf1* ablation on survival (S5 Fig).

As observed in the other cases, the double knockout did not improve the response profile compared to *Raf1* deletion alone. Although we saw a slightly higher percentage of CR in the double knockout group (11.6% vs. 8.4%), the PR rate was lower (29% vs. 52.4%). The proportion of PD and SD was also higher (39.1% and 20.3%, respectively, vs. 23.8% and 15.4%) (Fig 5A–5B). Overall, there was no evidence of an additive or synergistic benefit from DDR1 co-targeting. The total tumor burden reduction was not significantly different from that achieved with *Raf1* deletion alone (Fig 5C).

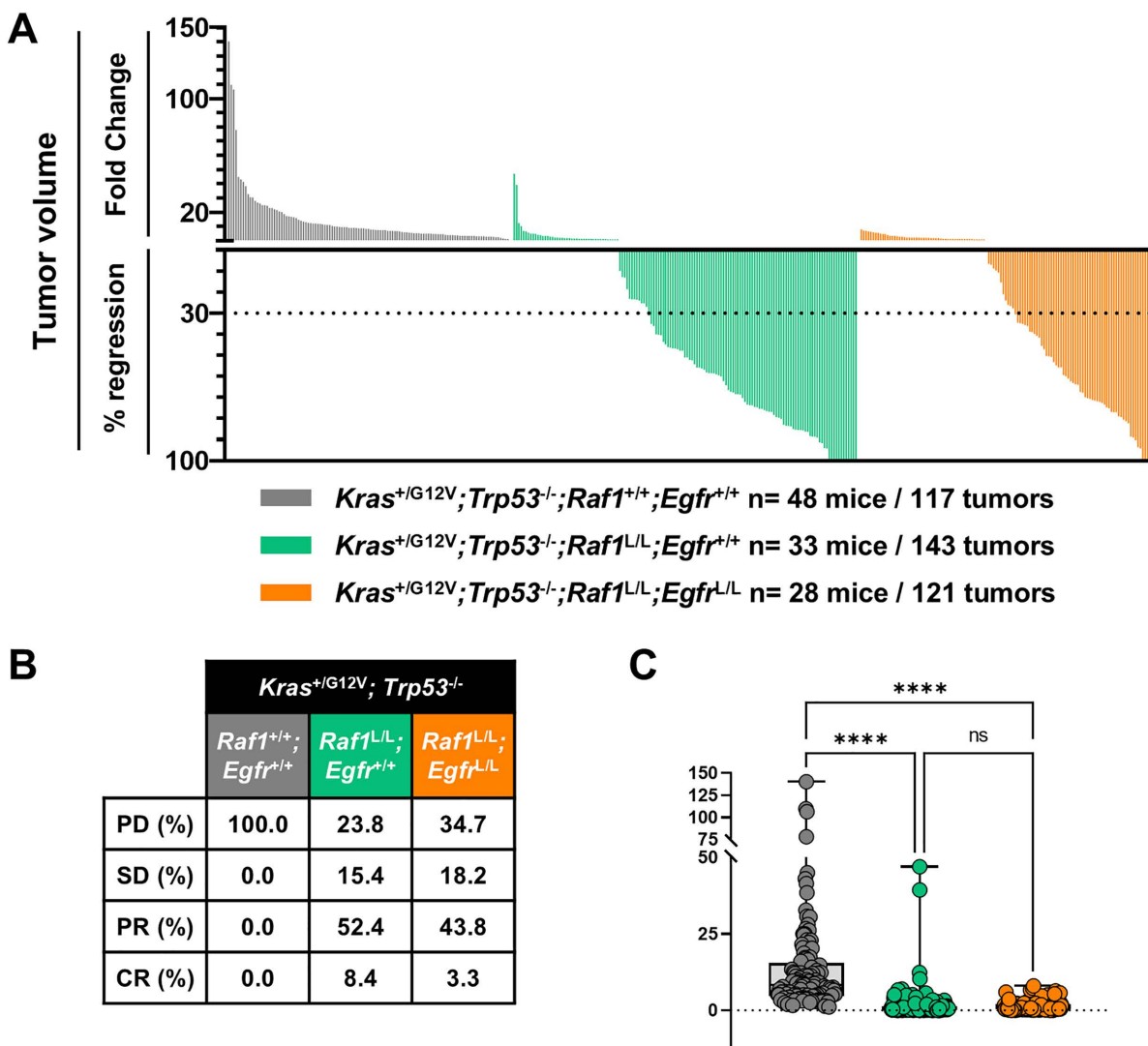

**Fig 4. Effect of *Egfr* and *Raf1* genetic ablation on LUAD development. (A)** Waterfall plot representing the FC in tumor volume and the percentage of regression for individual CT-positive lung tumors in *Kras⁺/ᶠˢᶠᴳ¹²ⱽ;Trp53ᶠ/ᶠ;hUBC-CreERT2⁺/ᵀ;Raf1⁺/⁺;Egfr⁺/⁺* (grey), *Kras⁺/ᶠˢᶠᴳ¹²ⱽ;Trp53ᶠ/ᶠ;hUBC-CreERT2⁺/ᵀ; Raf1ᴸ/ᴸ;Egfr⁺/⁺* (green), and *Kras⁺/ᶠˢᶠᴳ¹²ⱽ;Trp53ᶠ/ᶠ;hUBC-CreERT2⁺/ᵀ;Raf1ᴸ/ᴸ;Egfrᴸ/ᴸ* (orange) mice following 2 months of TMX diet. **(B)** Table indicating the percentage of tumors that show PD, SD, PR, and CR. **(C)** Statistical comparison of tumor volume FC between groups. *p*-values were obtained using the Kruskal-Wallis test with multiple comparisons. Data are shown as in Fig 3.

These findings suggest that, while DDR1 may contribute to early stages of tumorigenesis, it is not required for tumor maintenance once RAF1 is removed. Unlike in the initiation setting, co-targeting DDR1 does not enhance the therapeutic outcome in established KRAS-driven LUAD, reinforcing RAF1 as the essential and sufficient vulnerability in this context.

Across three independent genetic models, concomitant deletion of *Raf1* with other candidate therapeutic targets failed to improve significantly tumor regression beyond *Raf1* ablation alone. Dual targeting of *Araf*, *Egfr*, or *Ddr1* did not significantly reduce tumor burden or increase the frequency of partial or complete responses compared with single *Raf1* deletion (Fig 6). While *Raf1* elimination consistently drove tumor regression, co-targeting additional kinases provided no additive or synergistic benefit. Importantly, none of the dual deletion strategies increased systemic toxicity, as evidenced by stable body weight, absence of overt tissue abnormalities, and unaltered survival (Fig 2, S3 Fig, S5

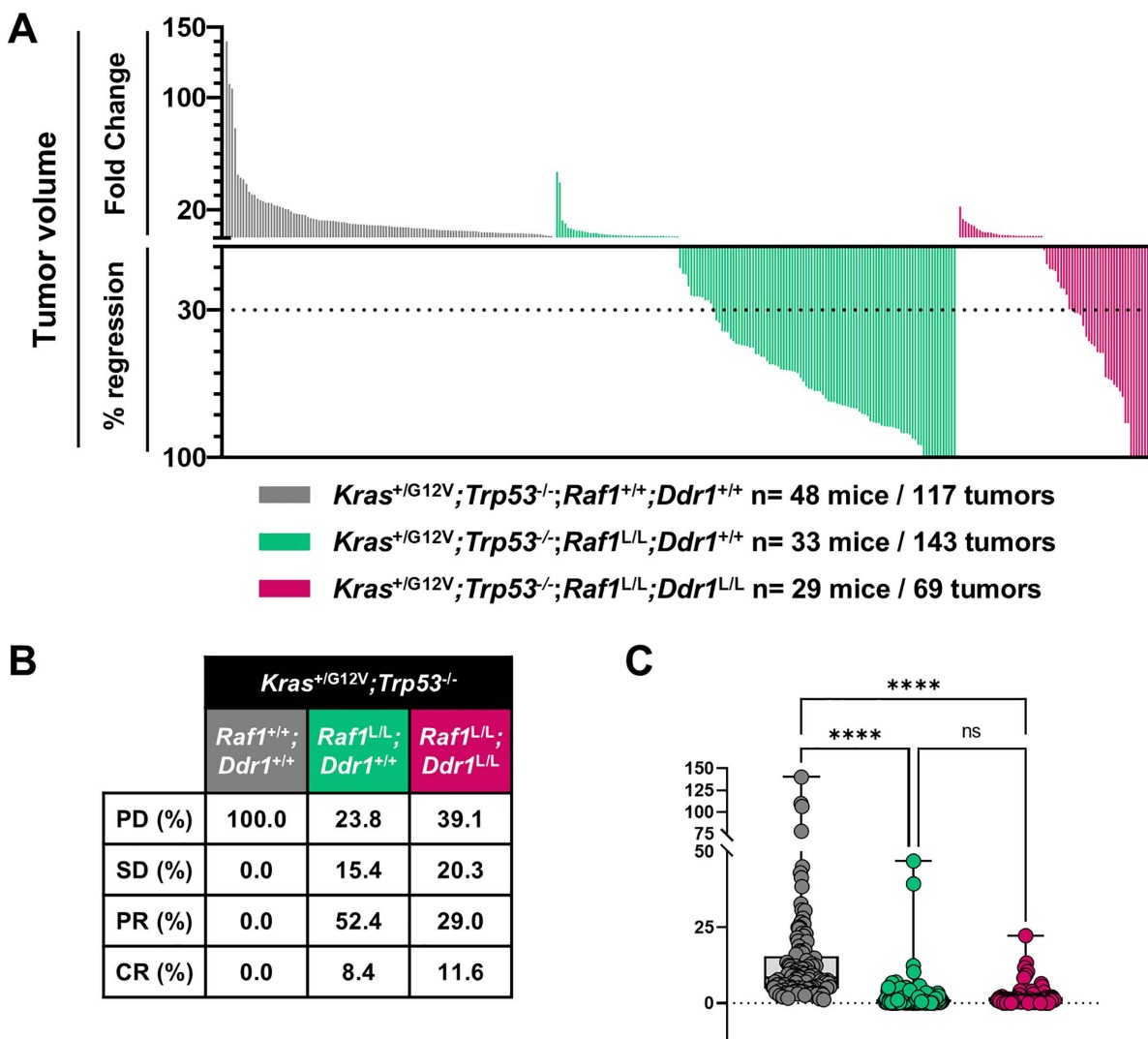

**Fig 5. Effect of *Ddr1* and *Raf1* genetic ablation on LUAD development. (A)** Waterfall plot representing the FC in tumor volume and the percentage of regression for individual CT-positive lung tumors in *Kras*[+/FSFG12V];*Trp53*[F/F];*hUBC-CreERT2*[+/T];*Raf1*[+/+];*Ddr1*[+/+] (grey), *Kras*[+/FSFG12V]; *Trp53*[F/F];*hUBC-CreERT2*[+/T];*Raf1*[L/L];*Ddr1*[+/+] (green), and *Kras*[+/FSFG12V];*Trp53*[F/F];*hUBC-CreERT2*[+/T];*Raf1*[L/L];*Ddr1*[L/L] (pink) mice following 2 months of TMX diet. **(B)** Table indicating the percentage of tumors that show PD, SD, PR, and CR. **(C)** Statistical comparison of tumor volume FC between groups. *p*-values were obtained using the Kruskal-Wallis test with multiple comparisons. Data are shown as in Fig 3.

Fig, S7 Fig). These data consolidate RAF1 as the key vulnerability in KRAS-driven LUAD and suggest that eliminating RAF1 alone is sufficient to achieve maximal therapeutic benefit. Furthermore, the lack of adverse effects observed upon *Araf* co-deletion supports the anticipated safety of future RAF1-directed degraders with partial cross-reactivity towards ARAF.

## Discussion

The therapeutic landscape for KRAS-mutant LUAD remains limited, particularly for tumors harboring non-G12C alleles or those that develop resistance to KRAS[G12C] inhibitors. In this context, targeting key downstream effectors of KRAS

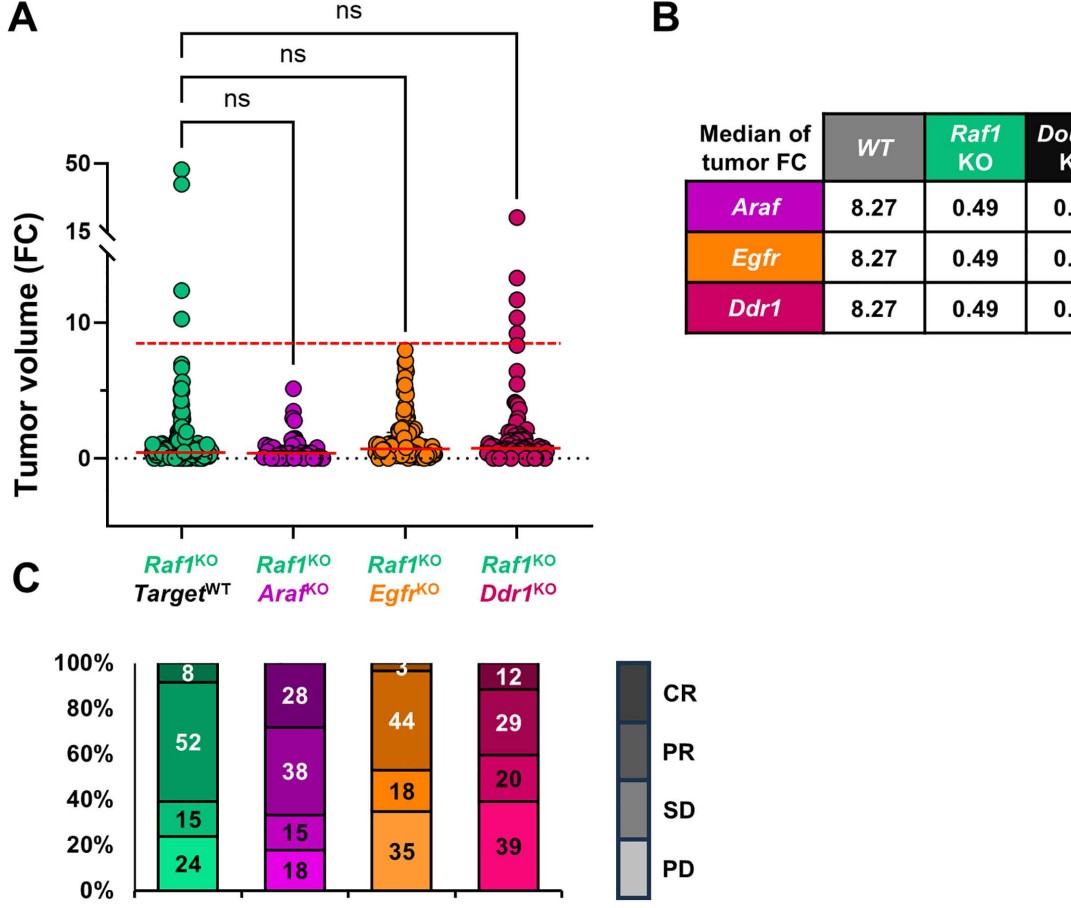

**Fig 6. Comparison of tumor responses between *Raf1* single ablation and double knockouts models. (A)** FC in tumor volume after 60 days of treatment in mice with systemic deletion of *Raf1* alone (green) or in combination with *Araf* (purple), *Egfr* (orange), or *Ddr1* (pink). Each dot represents an individual tumor. Horizontal red solid lines indicate the median tumor volume for each group; dashed red lines indicate the reference median for the WT (*Raf1*[+/+];*Target*[+/+]) control. Statistical analysis was performed using Kruskal-Wallis test with multiple comparisons. **(B)** Table showing median tumor volume FC for each genotype. **(C)** Stacked bar plots indicating the percentage of tumors classified as CR, PR, SD, or PD according to RECIST-like criteria.

signaling emerges as a promising strategy. Among them, RAF1 has gained attention as a functionally indispensable mediator of tumor maintenance.

Our findings confirm previous studies showing that RAF1 ablation leads to significant tumor regression in LUAD models driven by KRAS[G12V] and p53 loss [6,21]. Unlike traditional kinase inhibitors, genetic deletion removes both enzymatic and scaffold functions, many of which are critical for tumor cell survival and the suppression of apoptosis. This is particularly relevant for RAF1, as prior GEMM studies demonstrated that substituting the WT allele with kinase-dead variants (RAF1[K375M] and RAF1[D468A]) failed to reproduce the antitumor effects of complete RAF1 loss [7]. These observations make RAF1 an ideal candidate for targeted protein degradation strategies, which can uniquely eliminate non-enzymatic functions that are inaccessible to conventional small-molecule inhibitors.

In this work, we used a dual-recombinase GEMM platform to model therapeutic intervention in established LUAD and evaluated whether concomitant deletion of *Raf1* with *Araf*, *Egfr*, or *Ddr1* could enhance tumor regression or introduce toxicity in adult mice with established disease. Again, *Raf1* deletion consistently produced robust antitumor responses, as previously published [6,21], while co-targeting of additional RAF isoforms or upstream RTKs failed to improve therapeutic outcomes. Importantly,

none of the dual-targeting strategies resulted in adverse systemic effects, supporting the safety of targeting RAF1 alone, even in combination with other structurally related proteins, such as ARAF. These results further establish RAF1 as a non-redundant mediator of tumor maintenance in KRAS-driven LUAD and confirm its unique role within the RAF kinase family.

The three members of the RAF kinase family—ARAF, BRAF, and RAF1—share a high degree of sequence homology and conserved structural organization, with key residues preserved across all isoforms. This similarity has historically hindered the development of isoform-selective inhibitors, leading instead to the generation of pan-RAF compounds. However, several studies have revealed isoform-specific differences in biological functions, regulation, and signaling output. One relevant distinction concerns their differential interaction with the HSP90-CDC37 chaperone complex. RAF1 is a well-established constitutive client of this system, whereas BRAF associates with it only under specific conditions, such as in the presence of the oncogenic BRAF[V600E] mutation [22,23]. ARAF has also been identified as a client of this chaperone machinery [24,25], raising the possibility that ARAF may undergo co-degradation in RAF1-targeting strategies, a consideration of translational importance for degrader design.

Genetic studies underscore further functional divergence among the isoforms. While *Braf* or *Raf1* knockout mice exhibit embryonic lethality due to developmental defects—vascular and neuronal in the case of BRAF, massive hepatic apoptosis in the case of RAF1—*Araf*-null mice survive to term but die shortly after birth from neurological and gastrointestinal abnormalities [26–29]. These findings highlight that, although all three proteins regulate ERK pathway signaling, they execute non-redundant, isoform-specific functions.

Compared to RAF1, whose essential role in both tumor initiation and maintenance in KRAS-driven cancer is well established [12,30], the physiological and pathological relevance of ARAF remains poorly defined. With the lowest basal kinase activity of the family (~20% of RAF1; [10]), ARAF's biological contribution may rely on kinase-independent functions. For example, it can inhibit apoptosis during epithelial differentiation by directly binding to MST2, mimicking RAF1's anti-apoptotic scaffold role [31,32]. Splice variants lacking kinase activity have also been reported, acting as dominant-negative regulators or competitive scaffolds that can interfere with RAF dimerization and MEK phosphorylation, thereby attenuating ERK1/2 activation [11]. Emerging evidence suggests that ARAF can acquire context-specific oncogenic or tumor-suppressive functions. Genomic studies have identified *Araf* copy number gains and activating mutations in LUAD as potential predictors of response to sorafenib [33], while the gain-of-function mutation ARAF[S214P] has been linked to trametinib sensitivity in anomalous lymphatic disease [34]. Conversely, Mooz *et al.* have described a non-canonical, kinase-independent tumor-suppressive role for ARAF in NSCLC, whereby it limits ERBB3 expression and PI3K/AKT signaling, reducing metastatic potential [35]. However, these observations were made in non-KRAS-driven models, underscoring the importance of oncogenic context in defining ARAF's role.

In our study, we directly addressed ARAF's contribution to KRAS-driven LUAD using both initiation and therapeutic models. In the initiation setting, concurrent deletion of *Araf* and activation of *Kras*[G12V] did not alter tumor burden or histological grade after six months, indicating that ARAF is not required for tumor formation in this context. In the therapeutic setting, deletion of *Araf* alone in established *Kras*[G12V];*Trp53*[-/-] LUAD lesions had no effect on tumor growth. Moreover, concomitant deletion of *Araf* and *Raf1* failed to enhance the robust regression induced by *Raf1* ablation alone. These findings position ARAF alongside BRAF as dispensable for tumor maintenance in KRAS-driven LUAD, in contrast to the essential role of RAF1.

From a translational perspective, the lack of toxicity observed upon systemic ARAF deletion—alone or in combination with RAF1—has direct implications for drug development. The clinical development of pan-RAF inhibitors, such as tovorafenib or naporafenib, has aimed to overcome the limitations of early BRAF-selective inhibitors (e.g., paradoxical activation) by targeting the kinase activity of all RAF isoforms, including ARAF. However, concerns regarding the cumulative toxicity of inhibiting multiple members of the MAPK pathway remain. Given the structural similarity between RAF1 and ARAF, degraders designed to eliminate RAF1 may also partially target ARAF. Our results suggest that such off-target co-degradation—or pan-RAF kinase inhibition—is unlikely to cause adverse effects and should not be a barrier to the clinical advancement of RAF1-directed degradation strategies.

Similarly, DDR1 was prioritized based on its reported role in KRAS-driven LUAD progression and its unique biology as a collagen-activated receptor tyrosine kinase. Ambrogio *et al.* demonstrated that genetic deletion of *Ddr1* or pharmacological inhibition with 7rh impaired tumor initiation and slowed progression in LUAD GEMMs, particularly when combined with blockade of the NOTCH pathway. These effects were attributed to DDR1's ability to remodel the tumor microenvironment, regulate extracellular matrix interactions, and sustain proliferative signaling [14]. In our study, however, co-deletion of *Ddr1* and *Raf1* in established tumors did not enhance regression compared to *Raf1* loss alone. In fact, the overall response rate was slightly lower in the dual-knockout cohort, suggesting that DDR1's contribution is more critical in early tumorigenesis than in maintaining advanced lesions.

Together, these results highlight that the therapeutic relevance of co-targeting upstream or parallel pathways is highly context- and stage-dependent. For established KRAS-driven LUAD, RAF1 ablation appears to eliminate the need for EGFR- and DDR1-mediated inputs, underscoring its sufficiency as a single target. A relevant question is whether the essential role of RAF1 is restricted to the KRAS$^{G12V}$ mutation used in this study. Although our results are based on this specific model, preliminary evidence from our laboratory using KRAS$^{G12C}$ and KRAS$^{G12D}$ GEMMs suggests that the dependence on RAF1 is a conserved feature across the most prevalent KRAS-mutant isoforms in LUAD. Further studies will be required to confirm if this also applies to less frequent mutations such as G13 or Q61.

Our study provides strong preclinical rationale for the development of RAF1 degraders as therapies in KRAS-mutant LUAD. Given that its tumor-promoting functions are largely non-catalytic, TPD technologies such as Proteolysis Targeting Chimeras (PROTACs) or molecular glues are ideally suited to eliminate both kinase-dependent and kinase-independent RAF1 functions. Importantly, our data show that even in the absence of absolute isoform selectivity, such agents are unlikely to induce toxicity through ARAF co-degradation. This finding broadens the chemical space available for degrader design and alleviates safety concerns about collateral activity on closely related RAF kinases.

In addition to confirming the sufficiency of RAF1 elimination, our study also provides a valuable result: the absence of added benefit from combinatorial targeting strategies involving EGFR or DDR1, at least in the KRAS mutant LUAD context. These findings help refine the list of promising co-targets and suggest that future combinatorial approaches should focus on other signaling nodes. In this regard, while RAF1 ablation represents a potent therapeutic intervention, achieving complete and permanent tumor eradication may require vertical inhibition of the pathway. Preliminary evidence from our laboratory using GEMM models for other KRAS alleles suggests that combining RAF1 systemic depletion with direct KRAS inhibitors (e.g., G12C or G12D inhibitors) could provide the necessary synergy to overcome adaptive resistance and induce more durable clinical responses. Thus, our study not only establishes RAF1 as a standalone target but also defines the boundaries for future rational combinations in the era of direct KRAS targeting.

## Conclusions

Our findings confirm that RAF1 ablation alone is sufficient to induce significant tumor regression in established KRAS-driven LUAD and that co-targeting ARAF, EGFR, or DDR1 provides no added therapeutic benefit. This study validates RAF1 as a central vulnerability in LUAD and supports the development of selective or partially selective RAF1 degraders as a monotherapy approach. The absence of systemic toxicity upon *Araf* co-deletion further reinforces the potential safety of degraders with non-selective RAF1 activity. Together, our results offer a robust preclinical framework for the translation of RAF1-targeted degradation into clinical therapies for KRAS-mutant lung cancer.

## Supporting information

**S1 Fig. Generation of an *Araf*- conditional knockout allele by gene targeting.** (A) Schematic representation of the wild-type *Araf* locus, the targeting vector and the targeted allele. Black boxes, *Araf* exons; Neo, PGK-Neomycine cassette; white diamond, FRT site; black triangle, loxP site. The positions of the probes (5' and 3'), and restriction enzyme sites

(B, BamHI) are indicated. The expected diagnostic fragments following BamHI digestion are represented by arrows. (B) Southern blot analysis of gDNA from recombinant ES cell clones carrying the recombinant *Araf* allele.
(TIF)

**S2 Fig. Therapeutic mouse model for assessment of tumor response to ablation of selected targets.** Schematic representation of gene expression and ablation during lung adenocarcinoma tumor development. The $Kras^{FSFG12V}$ and $Trp53^F$ alleles undergo recombination following Ad5-CMV-FLPo infection. The $hUBC\text{-}CreERT2^T$, $Raf1^L$, and the additional targeted alleles are either activated or recombined upon exposure to TMX. Dark colored boxes represent gene expression, whereas light colored boxes correspond to ablated genes.
(TIF)

**S3 Fig. *Egfr* and *Raf1* dual ablation does not affect body weight or survival.** (A) Body weight measurements (in grams) of mice exposed to TMX-diet over time. Each point represents the mean weight of a group of mice, with error bars indicating the standard error of the mean (SEM). The gray points represent $Raf1^{+/+};Egfr^{+/+}$ mice (n = 60), while the orange points represent $Raf1^{L/L};Egfr^{L/L}$ mice (n = 19). (B) Kaplan-Meier survival curves representing data from $Raf1^{+/+};Egfr^{+/+}$ (n = 62, grey), $Raf1^{L/L};Egfr^{L/L}$ (n = 8, orange) mice.
(TIF)

**S4 Fig. Effect of *Raf1* deletion and afatinib treatment on LUAD development.** (A) Waterfall plot representing the FC in tumor volume and the percentage of regression for individual CT-positive lung tumors in $Kras^{+/FSFG12V};Trp\text{-}53^{F/F};hUBC\text{-}CreERT2^{+/T};Raf1^{+/+}$ (grey), $Kras^{+/FSFG12V};Trp53^{F/F};hUBC\text{-}CreERT2^{+/T};Raf1^{L/L}$ (green), and $Kras^{+/FSFG12V};Trp\text{-}53^{F/F};hUBC\text{-}CreERT2^{+/T};Raf1^{L/L}$ + afatinib (mustard) mice following 2 months of treatment. (B) Table indicating the percentage of tumors that show PR, CR, PD, and SD. (C) Statistical comparison of tumor volume FC between groups. p-values were obtained using the Kruskal-Wallis test with multiple comparisons. Data are shown as in Fig 3.
(TIF)

**S5 Fig. *Ddr1* and *Raf1* dual ablation does not affect body weight or survival.** (A) Body weight measurements (in grams) of mice exposed to TMX-diet over time. Each point represents the mean weight of a group of mice, with error bars indicating the standard error of the mean (SEM). The gray points represent $Raf1^{+/+};Ddr1^{+/+}$ mice (n = 60), while the pink points represent $Raf1^{L/L};Ddr1^{L/L}$ mice (n = 36). (B) Kaplan-Meier survival curves representing data from $Raf1^{+/+};Ddr1^{+/+}$ (n = 62, grey), $Raf1^{L/L};Ddr1^{L/L}$ (n = 47, pink) mice.
(TIF)

**S6 Fig. Validation of RAF1, DDR1, EGFR, and ARAF knockdown in generated models.** (A) Systemic *Raf1* and *Araf* ablation confirmed in representative tissues. Upper panel, PCR analysis of *Raf1* ablation; lower panel, PCR analysis of *Araf* ablation. Lanes include WT, lox control, experimental samples, and a water negative control. (B–C) Representative immunostaining of paraffin-embedded lung tumor sections showing α-RAF1 in combination with α-EGFR (B) or α-DDR1 (C). Scale bars: 200 μm.
(TIF)

**S7 Fig. Kaplan–Meier survival analysis of all tumor-bearing mice under TMX treatment.** (A) Survival curves representing data from wild-type $Raf1^{+/+}$ (n = 62, grey), $Raf1^{L/L}$ (n = 34, green), $Raf1^{L/L};Araf^{L/Y}$ (n = 28, purple), $Raf1^{L/L};Ddr1^{L/L}$ (n = 47, pink), $Raf1^{L/L};Egfr^{L/L}$ (n = 8, orange) mice. (B) Log-rank (Mantel–Cox) test comparing *Raf1* single ablation with Raf1 ablation in combination with a second target.
(TIF)

S1 Table. **Description of the modified alleles present in the compound mouse strains.** Capital 'L', 'F', and 'T' stand for 'loxP', 'FRT', or 'transgene' respectively.
(DOCX)

S1 Minimal Data Set. **Raw numerical values used to generate all graphs and plots presented in both the main manuscript and the supporting information files.** The data are organized by figure and include individual data points.
(XLSX)

S1 Raw Figures. **Original graphical representations with embedded GraphPad Prism objects for all figures in the study.** Each object contains the underlying source data and statistical analyses, allowing for full transparency and replication of the experimental results.
(PPTX)

## Acknowledgments

The authors would like to express their gratitude to Marta San Román, Alejandra López-García, and Raquel Villar for excellent technical assistance; Isabel Blanco (Animal Facility) and Francisca Mulero (Molecular Imaging Unit) of CNIO for their technical support.

## Author contributions

**Conceptualization:** Mónica Musteanu, Mariano Barbacid, Sara García-Alonso.

**Data curation:** Laura de-la-Puente-Ovejero, Ana Fernández-Rodríguez, Sarah Francoz, Sara García-Alonso.

**Formal analysis:** Laura de-la-Puente-Ovejero, Ana Fernández-Rodríguez, Sarah Francoz, Sara García-Alonso.

**Funding acquisition:** Carmen Guerra, Mónica Musteanu, Mariano Barbacid.

**Methodology:** Laura de-la-Puente-Ovejero, Ana Fernández-Rodríguez, Sarah Francoz, Gonzalo Aizpurua, Lucía Lomba-Riego, Sara García-Alonso.

**Project administration:** Mónica Musteanu, Mariano Barbacid, Sara García-Alonso.

**Resources:** Matthias Drosten, Carmen Guerra.

**Supervision:** Mónica Musteanu, Sara García-Alonso.

**Writing – original draft:** Laura de-la-Puente-Ovejero, Sara García-Alonso.

**Writing – review & editing:** Laura de-la-Puente-Ovejero, Ana Fernández-Rodríguez, Gonzalo Aizpurua, Mónica Musteanu, Mariano Barbacid, Sara García-Alonso.

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
