## [Decision Letter · Decision Letter 0]

17 Dec 2025

Dear Dr. Alonso,

Thank you for submitting your manuscript to PLOS ONE. After careful consideration, we feel that it has merit but does not fully meet PLOS ONE’s publication criteria as it currently stands. Therefore, we invite you to submit a revised version of the manuscript that addresses the points raised during the review process.

We look forward to receiving your revised manuscript.

Kind regards,

Marco Trerotola

Academic Editor

PLOS One

Journal Requirements:

2. To comply with PLOS One submission requirements, in your Methods section, please provide additional information regarding the experiments involving animals and ensure you have included details on methods of sacrifice.

“This research was funded by the Agencia Estatal de Investigación, Ministerio de Ciencia, Innovación y Universidades (MICIU/AEI/10.13039/501100011033: RTI2018-094664-B-I00, PID2021-122797OB-I00) cofounded by “ERDF A way of making Europe”, the Autonomous Community of Madrid (S2022/BMD-7437, iLUNG 2.0-CM), and the CRIS Cancer Foundation. M.B. is a recipient of an Endowed Chair from the AXA Research Fund. M.B. is a recipient of a CIBERONC Fund (CB21/12/00121). S.G-A. was partially supported (from 2020 until 2022) by AECC Postdoctoral Fellowship (call 2020). S.G-A was partially supported (from 2019 until 2020) by the Juan de la Cierva Investigadores fellowship (FJC2018-036013-I), M.M. is recipient of a Ramón y Cajal fellowship (RYC2018-025415-I), L.d-l-P-O. and A.F-R. are supported by the predoctoral fellowships (FPU21/04678 and FPU22/02924, respectively) all funded by Agencia Estatal de Investigación, Ministerio de Ciencia, Innovación y Universidades (MICIU/AEI/10.13039/501100011033) and cofounded by “ESF Investing in your future”. G.A. and L.L-R. were supported by a fellowship from “La Caixa” Foundation (LCF/BQ/DR22/11950011 and LCF/BQ/DR23/12000028, respectively)."

“This research was funded by the Agencia Estatal de Investigación, Ministerio de Ciencia, Innovación y Universidades (MICIU/AEI/10.13039/501100011033: RTI2018-094664-B-I00, PID2021-122797OB-I00) cofounded by “ERDF A way of making Europe”, the Autonomous Community of Madrid (S2022/BMD-7437, iLUNG 2.0-CM), and the CRIS Cancer Foundation. M.B. is a recipient of an Endowed Chair from the AXA Research Fund. M.B. is a recipient of a CIBERONC Fund (CB21/12/00121). S.G-A. was partially supported (from 2020 until 2022) by AECC Postdoctoral Fellowship (call 2020). S.G-A was partially supported (from 2019 until 2020) by the Juan de la Cierva Investigadores fellowship (FJC2018-036013-I), M.M. is recipient of a Ramón y Cajal fellowship (RYC2018-025415-I), L.d-l-P-O. and A.F-R. are supported by the predoctoral fellowships (FPU21/04678 and FPU22/02924, respectively) all funded by Agencia Estatal de Investigación, Ministerio de Ciencia, Innovación y Universidades (MICIU/AEI/10.13039/501100011033) and cofounded by “ESF Investing in your future”. G.A. and L.L-R. were supported by a fellowship from “La Caixa” Foundation (LCF/BQ/DR22/11950011 and LCF/BQ/DR23/12000028, respectively)."

5. We note that your Data Availability Statement is currently as follows: [All relevant data are within the manuscript and its Supporting Information files.]

Reviewers' comments:

Reviewer's Responses to Questions

**Comments to the Author**

1. Is the manuscript technically sound, and do the data support the conclusions?

Reviewer #1: Yes

Reviewer #2: Partly

2. Has the statistical analysis been performed appropriately and rigorously?

Reviewer #1: Yes

Reviewer #2: Yes

3. Have the authors made all data underlying the findings in their manuscript fully available?

Reviewer #1: Yes

Reviewer #2: Yes

4. Is the manuscript presented in an intelligible fashion and written in standard English?

Reviewer #1: Yes

Reviewer #2: Yes

Reviewer #1: Review Summary

This study provides further insight into the role of c-RAF/RAF1 as a therapeutic vulnerability in RAS-driven cancers, where-by deletion of c-RAF/RAF1 in vivo was evaluated in combination with deletion of ARAF, EGFR or DDR1 to determine potential synergistic/additive co-targeting strategies in KRAS-driven lung adenocarcinoma. It was found that ARAF, EGFR nor DDR1 co-deletion provided no additional therapeutic benefit, and no additive systemic toxicity was observed (suggesting therapies co-targeting c-RAF/RAF1 and ARAF are likely to be safe). More, EGFR co-deletion was previously found to be synergistic in KRAS-driven pancreatic ductal adenocarcinoma mouse models, however this was not found to occur in the lung adenocarcinoma mouse models evaluated in this study, suggesting a potential cancer lineage/tissue-specific context (and/or broader mutational profile context). This study effectively presents negative data that will enable on-going and future therapeutic strategies aimed at neutralising c-RAF (particularly its kinase-independent functions and/or via targeted degradation) in RAS-mutant cancers.

Minor Corrections

I would recommend acceptance of this manuscript upon actioning the following minor corrections:

Inclusion of discussive point(s) considering the translatability of these findings against other KRAS mutations (beyond the sole G12V model tested in this study), G12C/D, G13, Q61.

Inclusion of discussive point(s) considering rationale combination strategies that may confer additive/synergistic action in this model (and models like it), as even a c-RAF/RAF1 targeted degrader is unlikely to induce durable tumour regression as a monotherapy.

Evidence of ARAF, DDR1 and EGFR knockdown at the protein level (e.g., via western immunoblotting) is needed for confirmation of model generation. Supplementary figure is appropriate in this case. Same approach as is seen in: Blasco, et al. Cancer Cell. (2019) 35(4): 573-587

Reviewer #2: I had the pleasure of reviewing your article "RAF1 as a standalone therapeutic target in KRAS-driven adenocarcinoma" for PLOS one. Overall, the manuscript is well written. The experiments are well controlled and in general the data support the conclusions. The work is an important contribution to the literature that lays the groundwork for the possibility of using a RAF1-degrader as a therapeutic for KRAS mutated LUAD. I had several relatively minor concerns to be addressed that I believe would improve the manuscript

1) Could the methods for necropsy/histopathology be added? Is there a pathologist that was consulted for the work?

2) Why are there so few mice in the Raf1+/+ Araf L/Y group in Figure 3?

3) Could a KM analysis be added for all of the tumor bearing mice?

4) The conclusion that RAF1 is the dominant vulnerability in KRAS-driven LUAD might be a little bit overstated. What about KRAS itself?

5) Also overstated is that the findings in the manuscript demonstrate that RAF1 ablation alone is sufficient to induce significant tumor regression in established KRAS-driven LUAD as this finding has already been published. Perhaps change the verb demonstrate to confirm.

6) A discussion of the available "pan" RAF kinase inhibitors and their effects on the kinase activity of ARAF, and how that relates to their toxicity, might add translational context for the conclusions

Thank you to the editors for the opportunity to review this manuscript.

**Do you want your identity to be public for this peer review?** For information about this choice, including consent withdrawal, please see our Privacy Policy

Reviewer #1: **Yes:** Dr. Connor M. Blair

Reviewer #2: No

---

## [Author Response · Author response to Decision Letter 1]

8 Jan 2026

Dear Dr. Trerotola,

We would like to thank you and the reviewers for the constructive comments and the positive evaluation of our manuscript. We have carefully addressed all the points raised by the editorial office and the reviewers.

Below is a point-by-point response to the reviewers’ comments. All changes in the manuscript have been highlighted using "Track Changes" and the line numbers where these modifications were introduced are indicated throughout this letter.

Responses to Journal Requirements:

We have reviewed the manuscript to ensure it meets PLOS ONE’s style requirements, including file naming and formatting of the title page and main body.

2. To comply with PLOS One submission requirements, in your Methods section, please provide additional information regarding the experiments involving animals and ensure you have included details on methods of sacrifice.

We have updated the "Materials and Methods" section to comply with PLOS ONE requirements. We have explicitly stated the method of euthanasia (lines 138-139). In addition, as suggested by Reviewer #2, we have also included a new subsection titled Necropsy and Histopathology, providing a detailed description of the procedures used for tissue collection and analysis (lines 145-161).

3. Thank you for stating in your Funding Statement […]. Please provide an amended statement that declares *all* the funding or sources of support (whether external or internal to your organization) received during this study […]. Please also include the statement “There was no additional external funding received for this study.” in your updated Funding Statement. Please include your amended Funding Statement within your cover letter. We will change the online submission form on your behalf.

4. Thank you for stating the following financial disclosure […]. Please state what role the funders took in the study. If the funders had no role, please state: "The funders had no role in study design, data collection and analysis, decision to publish, or preparation of the manuscript." If this statement is not correct you must amend it as needed. Please include this amended Role of Funder statement in your cover letter; we will change the online submission form on your behalf.

We have updated our Funding Statement to include the required mandatory phrases.

5. We note that your Data Availability Statement is currently as follows: [All relevant data are within the manuscript and its Supporting Information files.] Please confirm at this time whether or not your submission contains all raw data required to replicate the results of your study […].

All raw data required (values behind means, individual data points for graphs) have been included as Supporting Information files (S1 Minimal Data Set and S1 Raw Figures).

6. If the reviewer’s comments include a recommendation to cite specific previously published works, please review and evaluate these publications to determine whether they are relevant and should be cited. There is no requirement to cite these works unless the editor has indicated otherwise.

Not applicable.

7. Please review your reference list to ensure that it is complete and correct. If you have cited papers that have been retracted, please include the rationale for doing so in the manuscript text, or remove these references and replace them with relevant current references […].

We have reviewed our reference list, and it is complete and correct.

Response to Reviewer #1

This study provides further insight into the role of c-RAF/RAF1 as a therapeutic vulnerability in RAS-driven cancers, where-by deletion of c-RAF/RAF1 in vivo was evaluated in combination with deletion of ARAF, EGFR or DDR1 to determine potential synergistic/additive co-targeting strategies in KRAS-driven lung adenocarcinoma. It was found that ARAF, EGFR nor DDR1 co-deletion provided no additional therapeutic benefit, and no additive systemic toxicity was observed (suggesting therapies co-targeting c-RAF/RAF1 and ARAF are likely to be safe). More, EGFR co-deletion was previously found to be synergistic in KRAS-driven pancreatic ductal adenocarcinoma mouse models, however this was not found to occur in the lung adenocarcinoma mouse models evaluated in this study, suggesting a potential cancer lineage/tissue-specific context (and/or broader mutational profile context). This study effectively presents negative data that will enable on-going and future therapeutic strategies aimed at neutralizing c-RAF (particularly its kinase-independent functions and/or via targeted degradation) in RAS-mutant cancers.

Minor Corrections

I would recommend acceptance of this manuscript upon actioning the following minor corrections:

1. Inclusion of discussive point(s) considering the translatability of these findings against other KRAS mutations (beyond the sole G12V model tested in this study), G12C/D, G13, Q61.

We agree that this is a relevant point. While our laboratory has traditionally focused on the KrasG12V model, we have recently expanded our research to include KrasG12C and KrasG12D GEMMs to evaluate allele-specific inhibitors. Our preliminary findings indicate that the requirement for RAF1 is consistent across these different oncogenic KRAS isoforms. We have added a paragraph in the Discussion addressing this translatability (Lines 495-500).

2. Inclusion of discussive point(s) considering rationale combination strategies that may confer additive/synergistic action in this model (and models like it), as even a c-RAF/RAF1 targeted degrader is unlikely to induce durable tumour regression as a monotherapy.

We agree with the reviewer that achieving durable, complete remissions in a clinical setting often requires combinatorial approaches to prevent the emergence of resistance. Our study demonstrates that co-targeting upstream receptors (EGFR, DDR1) or related kinases (ARAF) is not the optimal path in LUAD. Instead, our ongoing research (unpublished) suggests that vertical inhibition (combining RAF1 depletion with direct KRAS inhibitors) shows promising synergy. We have expanded the Discussion (Lines 514-521) to contextualize our findings within these future therapeutic strategies.

3. Evidence of ARAF, DDR1 and EGFR knockdown at the protein level (e.g., via western immunoblotting) is needed for confirmation of model generation. Supplementary figure is appropriate in this case. Same approach as is seen in: Blasco, et al. Cancer Cell. (2019) 35(4): 573-587

We have included a new S6 Fig to provide experimental confirmation of the ablation of our therapeutic targets. For EGFR, DDR1, and RAF1, we performed immunohistochemistry (IHC) on lung tumor sections from the relevant mouse cohorts. The results clearly demonstrate the absence of these proteins in the tumors upon genetic induction of Cre-recombinase. Regarding ARAF, despite testing several commercial options, we could not identify an antibody that provided reliable or specific staining for IHC or Western Blot in mouse lung tissue. To circumvent this, we have provided a PCR analysis from the same lung tissues, which demonstrates the efficient deletion of the Araf and Raf1 genes.

Furthermore, we have updated the "Tumor induction and tamoxifen exposure" subsection in the Materials and Methods (lines 198-199) to include a formal reference to these validation data.

Response to Reviewer #2

I had the pleasure of reviewing your article "RAF1 as a standalone therapeutic target in KRAS-driven adenocarcinoma" for PLOS one. Overall, the manuscript is well written. The experiments are well controlled and in general the data support the conclusions. The work is an important contribution to the literature that lays the groundwork for the possibility of using a RAF1-degrader as a therapeutic for KRAS mutated LUAD.

I had several relatively minor concerns to be addressed that I believe would improve the manuscript

1. Could the methods for necropsy/histopathology be added? Is there a pathologist that was consulted for the work?

We have added a detailed description of the necropsy and histopathology protocols in the Methods section (lines 145-161). We also confirm that the evaluation of the lesions was supervised by a trained pathologist, Dr. Eduardo Jose Caleiras, the head of the Histopathology Unit at CNIO.

2. Why are there so few mice in the Raf1+/+ Araf L/Y group in Figure 3?

We thank the reviewer for this observation. The Raf1+/+;ArafL/Y group (Figure 3) initially started with a cohort of 10 mice. However, the final number of mice represented in the tumor volume quantification was reduced to 4 due to the following technical and ethical reasons: (i) In some cases, the viral infection used for tumor induction caused localized pulmonary inflammation. This prevents accurate volumetric quantification of the tumors by CT, as the inflammatory infiltrate masks the tumor margins. To maintain the highest data rigor, these animals were excluded from the quantitative analysis; (ii) A subset of mice in this specific control group reached the human endpoints or died due to high tumor burden before the 60-day experimental completion, thus they could not be included in the final longitudinal comparison of tumor volume; (iii) Despite the reduction in the final “n” for quantification, we observed that ARAF ablation alone had no effect on tumor progression in any of the subjects. Consistent with the 3Rs principle and given the clear lack of phenotype, we decided that adding more animals to this control group was not ethically justified as it would not change the scientific conclusions of the study.

3. Could a KM analysis be added for all of the tumor bearing mice?

As suggested by the reviewer, we have generated a new Supplementary Figure (S7 Fig) that integrates the Kaplan-Meier survival curves for all tumor-bearing experimental groups (WT, Raf1 single ablation, and the double-ablation models for Araf, Egfr, and Ddr1). This unified analysis includes the respective “n” for each cohort and a comprehensive statistical comparison (Log-rank test). The results clearly show that the additional co-targeting of ARAF, EGFR, or DDR1 does not provide any further survival benefit. We have included a reference to this figure at the end of the Results section (line 382).

4. The conclusion that RAF1 is the dominant vulnerability in KRAS-driven LUAD might be a little bit overstated. What about KRAS itself?

We appreciate the reviewer’s comment and agree that KRAS remains the primary oncogenic driver and a major vulnerability in this type of cancer. Our intention was not to overlook the role of KRAS, but rather to emphasize that, among the various downstream effectors and co-targets evaluated in our study (ARAF, EGFR, and DDR1), RAF1 emerged as the most significant and non-redundant therapeutic vulnerability. To address this concern, we have revised the manuscript to use more nuanced terminology, replacing “dominant” or “primary” with terms such as “key” or “essential”. We have also ensured that the text correctly frames RAF1 as a fundamental mediator within the KRAS signaling landscape.

5. Also overstated is that the findings in the manuscript demonstrate that RAF1 ablation alone is sufficient to induce significant tumor regression in established KRAS-driven LUAD as this finding has already been published. Perhaps change the verb demonstrate to confirm.

We thank the reviewer for this suggestion. We have revised the manuscript throughout to replace “demonstrate” with “confirm” when referring to the effects of RAF1 ablation on tumor regression.

6. A discussion of the available "pan" RAF kinase inhibitors and their effects on the kinase activity of ARAF, and how that relates to their toxicity, might add translational context for the conclusions.

We agree that providing translational context regarding pan-RAF inhibitors strengthens the manuscript. As suggested, we have added a dedicated paragraph in the Discussion (lines 466-472) addressing this point. We discuss how current pan-RAF inhibitors in clinical development have shown clinical challenges related to toxicity and paradoxical activation. Our findings are particularly relevant here, as they demonstrate that even the complete genetic ablation of ARAF (which goes beyond kinase inhibition) does not increase systemic toxicity when combined with RAF1 loss.

---

## [Editor Report · Decision Letter 1]

12 Jan 2026

RAF1 as a standalone therapeutic target in KRAS-driven lung adenocarcinoma: no added efficacy from co-targeting ARAF, EGFR, or DDR1

PONE-D-25-51602R1

Dear Dr. Garcia-Alonso,

We’re pleased to inform you that your revised manuscript has been judged scientifically suitable for publication and will be formally accepted for publication once it meets all outstanding technical requirements.

Kind regards,

Marco Trerotola

Academic Editor

PLOS One

---

## [Editor Report · Acceptance letter]

PONE-D-25-51602R1

PLOS One

Dear Dr. García-Alonso,

I'm pleased to inform you that your manuscript has been deemed suitable for publication in PLOS One. Congratulations! Your manuscript is now being handed over to our production team.

Kind regards,

on behalf of

Professor Marco Trerotola

Academic Editor

PLOS One